# Evaluating the Therapeutic Potential of Curcumin and Synthetic Derivatives: A Computational Approach to Anti-Obesity Treatments

**DOI:** 10.3390/ijms25052603

**Published:** 2024-02-23

**Authors:** Marakiya T. Moetlediwa, Babalwa U. Jack, Sithandiwe E. Mazibuko-Mbeje, Carmen Pheiffer, Salam J. J. Titinchi, Elliasu Y. Salifu, Pritika Ramharack

**Affiliations:** 1Biomedical Research and Innovation Platform, South African Medical Research Council, Tygerberg 7505, South Africa; marakiya.moetlediwa@mrc.ac.za (M.T.M.); carmen.pheiffer@mrc.ac.za (C.P.); elliasu.salifu@mrc.ac.za (E.Y.S.); pritika.ramharack@mrc.ac.za (P.R.); 2Department of Biochemistry, North-West University, Mmabatho 2745, South Africa; sithandiwe.mazibukombeje@nwu.ac.za; 3Department of Obstetrics and Gynaecology, Faculty of Health Sciences, University of Pretoria, Pretoria 0001, South Africa; 4Department of Chemistry, Faculty of Natural Science, University of the Western Cape, Bellville 7535, South Africa; stitinchi@uwc.ac.za; 5Pharmaceutical Sciences, School of Health Sciences, University of KwaZulu-Natal, Westville, Durban 4001, South Africa

**Keywords:** pharmacokinetics, pharmacodynamics, obesity, curcumin, synthetic curcumin derivatives

## Abstract

Natural compounds such as curcumin, a polyphenolic compound derived from the rhizome of turmeric, have gathered remarkable scientific interest due to their diverse metabolic benefits including anti-obesity potential. However, curcumin faces challenges stemming from its unfavorable pharmacokinetic profile. To address this issue, synthetic curcumin derivatives aimed at enhancing the biological efficacy of curcumin have previously been developed. In silico modelling techniques have gained significant recognition in screening synthetic compounds as drug candidates. Therefore, the primary objective of this study was to assess the pharmacokinetic and pharmacodynamic characteristics of three synthetic derivatives of curcumin. This evaluation was conducted in comparison to curcumin, with a specific emphasis on examining their impact on adipogenesis, inflammation, and lipid metabolism as potential therapeutic targets of obesity mechanisms. In this study, predictive toxicity screening confirmed the safety of curcumin, with the curcumin derivatives demonstrating a safe profile based on their LD50 values. The synthetic curcumin derivative 1A8 exhibited inactivity across all selected toxicity endpoints. Furthermore, these compounds were deemed viable candidate drugs as they adhered to Lipinski’s rules and exhibited favorable metabolic profiles. Molecular docking studies revealed that both curcumin and its synthetic derivatives exhibited favorable binding scores, whilst molecular dynamic simulations showed stable binding with peroxisome proliferator-activated receptor gamma (PPARγ), csyclooxygenase-2 (COX2), and fatty acid synthase (FAS) proteins. The binding free energy calculations indicated that curcumin displayed potential as a strong regulator of PPARγ (−60.2 ± 0.4 kcal/mol) and FAS (−37.9 ± 0.3 kcal/mol), whereas 1A8 demonstrated robust binding affinity with COX2 (−64.9 ± 0.2 kcal/mol). In conclusion, the results from this study suggest that the three synthetic curcumin derivatives have similar molecular interactions to curcumin with selected biological targets. However, in vitro and in vivo experimental studies are recommended to validate these findings.

## 1. Introduction

Obesity has emerged as a public health threat, with drastic increases worldwide. According to the World Health Organization (WHO), about 13% of the global population are obese (body mass index [BMI] ≥ 30 kg/m^2^) and the prevalence of obesity is projected to increase to 20% by 2025 [1]. Obesity and overweight contribute to the rise of various metabolic complications such as inflammation [2], lipotoxicity [3], and dyslipidemia [4]. Also, the adverse effects of obesity and overweight contribute to the rise of non-communicable diseases such as diabetes [5], cardiovascular diseases (CVDs) [6], and different types of cancers [7]. Therefore, there is an urgent need for innovative therapeutic strategies to curb the prevalence of obesity and its underlying metabolic complications.

In recent years, natural products have gained considerable attention due to their potential health benefits [8,9]. Curcumin, a bioactive compound derived from the rhizome of the turmeric plant, otherwise called *Curcuma longa*, which belongs to the Zingiberaceae family, has shown promising anti-obesity properties through its ability to modulate various cellular pathways involved in lipid metabolism [10,11], adipogenesis [12,13,14], and inflammation [15,16]. While curcumin holds significant potential as an anti-obesity agent, its poor bioavailability and limited systemic exposure have hindered its clinical applications [17]. To address these challenges, studies have focused on the development of synthetic curcumin derivatives that are designed to improve the pharmacokinetic properties and enhance efficacy of curcumin. It is important to note that the cost of developing synthetic derivatives of curcumin depends on the complexity of the structure. In addition, the extraction of natural compounds is associated with a low yield and purity. As such, we previously reported that curcumin derivatives have the potential to ameliorate obesity and obesity-related metabolic complications even more effectively than curcumin [18]. 

The use of computational techniques in drug discovery and development has shown to be a useful tool for predicting the pharmacological properties of synthetic compounds, thus providing a cost-effective and time-efficient method of assessing their therapeutic potential [19]. Hence, in this study, we used computational techniques to explore the anti-obesity potential of curcumin and three previously synthesized curcumin derivatives. In a study by Ooko at al., (2016) [20], two chloro curcumin derivatives, specifically (1E,6E)-4-chloro-1,7-bis(3,4-dimethoxyphenyl)hepta-1,6-diene-3,5-dione (1A6) and (1E,6E)-4-chloro-1,7-bis(3-hydroxyphenyl)hepta-1,6-diene-3,5-dione (1A8), along with an asymmetric curcumin derivative, (1E,6E,8E)-1-(3,4-dimethoxyphenyl)-9-phenyldeca-1,6,8-triene-3,5-dione (1B8), were synthesized using a chemical reaction involving acetyl acetone or 3-chloroacetylacetone with boric oxide. Subsequently, the synthetic compounds underwent characterization through various spectroscopic techniques, including mass spectrometry (MS), infrared spectroscopy (FTIR), and proton (1H)-nuclear magnetic resonance (NMR) spectroscopy [20]. Moreover, these compounds demonstrated potential anti-cancer properties [20].

These findings lay the foundation to further evaluate the bioactivity of these synthetic curcumin derivatives, particularly for their anti-obesity potential. In this context, the present study aims to assess the pharmacokinetics and pharmacodynamics profiles of curcumin and synthetic curcumin derivatives (1A6, 1A8, and 1B8) using in silico approaches to identify and evaluate targeted obesity mechanisms. By leveraging computational models, we will evaluate their physicochemical properties, druglikeness, toxicity, and ADME properties, including absorption through the gastrointestinal tract, distribution to target tissues, and metabolism by hepatic enzymes. Furthermore, we explored the interactions between these compounds and their molecular targets involved in adipogenesis, lipid metabolism, and inflammation to predict their potential efficacy in combating obesity.

## 2. Results

### 2.1. Prediction of Toxicity for Curcumin and Synthetic Curcumin Derivatives

Certain medicinal compounds may pose toxic effects, causing harm to organs and cells. Such compounds have the potential to harm the liver, disrupt the immune system, cause cancer, and even become cytotoxic. Protox II was used to predict the toxicity of curcumin and three synthetic curcumin derivatives (1A6, 1A8, and 1B8), and the findings are summarized in Table 1. The results indicated that curcumin belongs to Toxicity Class 4 while the synthetic curcumin derivatives belong to Toxicity Class 5. Table 1 shows the lethal dose of 2000 mg/kg for curcumin, while 4000 mg/kg of 1A6 and 1B8 and 4400 mg/kg of 1A8 were predicted as lethal doses. Typically, drugs administered orally undergo a first-pass effect in the liver, which can be a primary factor in causing liver damage. Therefore, it is critical to assess the hepatotoxic effect of drugs. While curcumin, 1A6, and 1A8 were inactive for hepatoxicity at their lethal doses, 1B8 (4000 mg/kg) showed activity on hepatotoxicity (Table 1).

The carcinogenic potential of curcumin, 1A6, 1A8, and 1B8 was predicted, revealing that the compounds were inactive for carcinogenicity at their lethal doses (Table 1). This suggests a non-promotive effect on tumor development. Immunotoxicity, referring to the potential of drugs to induce an undesirable immune response, was also investigated. Curcumin, 1A6, and 1B8 exhibited activity towards immunotoxicity while 1A8 did not display immunotoxicity at their respective lethal doses. The compound 1B8 at 4000 mg/kg demonstrated cytotoxic effects, whereas curcumin, 1A6, and 1A8 exhibited non-cytotoxic profiles at their respective lethal doses.

### 2.2. Prediction of the Pharmacokinetic and Physicochemical Properties, Druglikeness, and Synthetic Scores of Curcumin and Synthetic Curcumin Derivatives

Pharmacokinetic and physicochemical properties are critical factors in determining the efficacy and safety of a drug. Apart from experimental models, determining the druglikeness of a compound remains crucial as it directly affects its absorption, metabolism, distribution, and elimination within the human system [21]. Lipinski’s rules (LRo5) are widely accepted as a guideline for predicting the ability of a compound to be a drug candidate. According to LRo5, a drug should have a molecular weight of 500 daltons or less, no more than 5 hydrogen bond donors, no more than 10 hydrogen bond acceptors, and an octanol–water partition coefficient value of 5 or less. However, it is important to note that LRo5 is not an absolute rule but should be used as a general guide. Other factors, such as the drug’s target and intended use, can also influence its druglikeness.

Physicochemical properties, such as lipophilicity, water solubility, molecular weight, and polarity, influence the absorption, distribution, metabolism, and elimination of a drug in the body [22]. For instance, lipophilic compounds readily dissolve in lipids. Such drugs are more easily absorbed across cell membranes and are usually metabolized at a slower pace than hydrophilic drugs [23]. As depicted in Table 2, the predicted lipophilic score (iLOGP) for curcumin, 1A6, 1A8, and 1B8 was 3.27, 3.88, 1.85, and 3.83, respectively. Among these compounds, 1A6 had the highest iLOGP, which suggests better cell membrane penetration in lipid-rich environments.

Typically, drugs with poor water solubility may be poorly absorbed and therefore have reduced bioavailability, while drugs with high polarity may have difficulty crossing cell membranes and reaching their intended target. Curcumin, 1A6, 1A8, and 1B8 were predicted to have poor solubility in water, based on their water solubility score being less than 0 mg/mL (Table 2). To understand the absorption and metabolism of curcumin, 1A6, 1A8, and 1B8, the GI absorption and the liver first-pass metabolism facilitated by the P-gp substrate, as well as cytochrome enzymes, were predicted. Curcumin, 1A6, 1A8, and 1B8 were predicted to have high GI absorbability and no inhibitory activity on the P-gp substrate (Table 2). All the compounds were predicted to inhibit the CYP3A4 and CYP2C9 enzymes, while 1A6 was predicted to inhibit CYP2D6, and the CYP1A2 enzyme was inhibited by 1A8 and 1B8 (Table 2).

Based on the LRo5 estimation, the results presented in Table 3 demonstrate that all compounds have a molecular weight below 500 daltons. Furthermore, it is observed that none of the compounds exceed the limit of 5 hydrogen bond donors or 10 hydrogen bond acceptors. Additionally, the octanol–water partition coefficient for all the compounds is less than 5. Therefore, these compounds satisfy the LRo5 criteria without any violations. The synthetic score of a compound typically ranges from 1 (easily synthesized) to 10 (difficult to synthesize). In this case, the compounds had a synthetic score lower than 4, indicating that they are relatively easy to synthesize.

### 2.3. Prediction of the Potential Biological Targets of Curcumin and Synthetic Curcumin Derivatives

We used two predictive webservers, SwissTarget prediction and SEA, and two online databases, PubMed and Google Scholar, to identify biological targets related to curcumin in obesity experimental models. This approach yielded a range of potential biological targets. For this study, we prioritized common biological targets consistently associated with curcumin across three key metabolic pathways: adipogenesis, lipid metabolism, and inflammation. We identified three common potential biological targets of curcumin in each pathway and compared them to three synthetic derivatives for molecular docking studies. Specifically, for adipogenesis, we selected peroxisome proliferator-activated receptor gamma (PPARγ), mitogen activated protein kinase 14 (MAPK14), and signal transducer and activator of ranscription 3 (STAT3) as potential biological targets. In the context of lipid metabolism, we focused on acetyl-CoA carboxylase (ACC), fatty acid synthase (FAS), and glycogen synthase kinase 3 beta (GSK3β) as the chosen potential targets. Similarly, for inflammation, our selection comprised cyclooxygenase 2 (COX2), interleukin 6 (IL6), and diponectinAdiponectin as potential biological targets. In the context of lipid metabolism, we focused on ACC, FAS, and GSK3β as the chosen potential targets. Similarly, for inflammation, our selection comprised COX2, IL6, and Adiponectin as potential biological targets.

### 2.4. Assessment of Protein-Ligand Binding Score by Molecular Docking and Post-Docking Analysis

#### 2.4.1. Assessing the Binding Affinities of Curcumin and Synthetic Curcumin Derivatives to Target Proteins Involved in Adipogenesis

Figure 1 illustrates the results of molecular docking experiments conducted on compounds targeting proteins involved in adipogenesis. The control standards used in these experiments were GW9662, LY2228820, and BB1608, established inhibitors of PPARγ, MAPK14, and STAT3, respectively. In comparison to GW9662 (−7.9 kcal/mol), the binding affinities of curcumin and synthetic curcumin derivatives (1A6, 1A8, and 1B8) to the PPARγ protein were found to be improved. Notably, the best docking pose was achieved by 1B8, which exhibited a docking score of −9.1 kcal/mol (Figure 1A). Following that, 1A8 displayed the second-best binding score to PPARγ with a docking score of −8.6 kcal/mol, followed by curcumin with a binding affinity of −8.4 kcal/mol. The compound 1A6 exhibited the lowest binding affinity among the docked compounds, with a docking score of −8.0 kcal/mol, although this was still higher than the control (Figure 1A).

LY2228820 served as the standard control for docking to MAPK14, yielding a docking score of −8.3 kcal/mol, and demonstrated better binding to MAPK14 compared to curcumin and its synthetic curcumin derivatives (Figure 1B). Specifically, 1A8 displayed the highest binding score of −7.3 kcal/mol compared to curcumin and its derivatives. Curcumin and 1B8 had docking scores of −6.6 kcal/mol and −7.0 kcal/mol, respectively, while 1A6 exhibited the poorest docking performance with a binding affinity of −6.5 kcal/mol (Figure 1B). BB1608 served as the standard control for docking to the STAT3 protein, with a binding score of −5.2 kcal/mol. The molecular docking of 1A6, 1B8, and curcumin outperformed the binding performance of the standard control. Specifically, 1A6 bound to STAT3 protein with a score of −5.8 kcal/mol, 1B8 had a binding affinity of −5.6 kcal/mol, and curcumin exhibited −5.5 kcal/mol. The lowest binding score was demonstrated by 1A8, which was −5.4 kcal/mol. Figure 1C depicts the docking scores of the compounds to STAT3 protein.

#### 2.4.2. Assessing the Binding Affinities of Curcumin and Synthetic Curcumin Derivatives to Target Proteins Involved in Inflammation

In Figure 2, the outcomes of molecular docking predictions are presented for compounds targeting proteins involved in inflammation. The docking predictions were performed using Naproxen, Raloxifene, and Rosiglitazone as reference controls, which are known ligands of COX2, IL6, and Adiponectin, respectively (Figure 2). For the docking of the COX2 protein, 1B8 (−9.0 kcal/mol) showed an improved binding affinity compared to Naproxen. The docking score for curcumin was the same as that of Naproxen at −8.9 kcal/mol. However, 1A6 displayed a lower binding affinity (−8.7 kcal/mol) compared to Naproxen, and 1A8 docked even more poorly to COX2 with a score of −8.5 kcal/mol compared to the other compounds (Figure 2A).

In the IL6 docking complex, 1A8 achieved the highest binding score of −5.9 kcal/mol, surpassing the reference control Raloxifene, which had a docking score of −5.7 kcal/mol (Figure 2B). Curcumin and 1B8 has the same binding score of −5.3 kcal/mol, while 1A6 exhibited a lower binding affinity of −5.2 kcal/mol (Figure 2B). When docking to Adiponectin, curcumin outperformed other compounds with a docking score of −8.1 kcal/mol. Rosiglitazone and 1B8 had similar binding affinities of −8.0 kcal/mol (Figure 2C). On the other hand, 1A6 demonstrated the lowest binding score of 7.4 kcal/mol in this case (Figure 2C).

#### 2.4.3. Assessing the Binding Affinities of Curcumin and Synthetic Curcumin Derivatives to Target Proteins Involved in Lipid Metabolism

Figure 3 depicts the outcomes of molecular docking predictions performed on compounds that target proteins involved in lipid metabolism. The docking predictions utilized PF05221304, Orlistat, and FT30356 as control standards of ACC, FAS, and GSK3β, respectively. When compared to PF05221304 with a −4.3 kcal/mol docking score, both curcumin and 1A8 demonstrated improved binding affinities to the ACC protein. Specifically, 1A8 exhibited the highest docking score of −5.2 kcal/mol, while curcumin achieved the second-highest binding affinity of −4.8 kcal/mol. The compounds 1B8 and 1A6 exhibited weaker binding to ACC compared to PF05221304, with binding affinities of −4.1 kcal/mol and −4.2 kcal/mol, respectively (Figure 3A).

In contrast to other compounds, Orlistat exhibited the weakest binding affinity to the FAS protein, with a docking score of −7.5 kcal/mol (Figure 3B). Molecular docking studies revealed that both 1A6 and 1A8 achieved the same docking scores of −10.0 kcal/mol, while 1B8 had a slightly lower binding score of −9.7 kcal/mol to the FAS protein (Figure 3B). Curcumin demonstrated the highest stability to the FAS protein, with a docking score of −10.5 kcal/mol (Figure 3B). For the GSK3β docking complex, the results showed that FT30356 had a binding score of −7.8 kcal/mol (Figure 3C). However, 1B8 surpassed the binding affinity of FT30356, with a docking score of −8.0 kcal/mol. Conversely, 1A6 displayed the weakest binding affinity at −7.4 kcal/mol (Figure 3C). Interestingly, curcumin and 1A8 exhibited the same binding affinities of −7.7 kcal/mol to the GSK3β protein (Figure 3C).

### 2.5. Assessment of Protein-Ligand Complex Stability and Conformational Flexibility by MD Simulations

#### 2.5.1. Assessing Structural Stability of PPARγ Protein in Its Interaction with Curcumin and Synthetic Curcumin Derivatives Using RMSD Analysis

Three target proteins, PPARγ, COX2, and FAS, that demonstrated the highest binding affinites in their respective metabolic pathways where further assessed for stabilies using DM simulations and binding free energies. This study evaluated the conformational changes in PPARγ in complex with GW9662, curcumin, 1A6, 1A8, and 1B8. The corresponding RMSD values were observed (Figure 4A). The obtained RMSD values shed light on the interactions of PPARγ with different ligands. Briefly, PPARγ without a ligand (APO) showed an RMSD value of 1.8 Å, suggesting that the protein itself was inherently stable during the simulation. This value serves as a baseline for comparing the structural dynamics of PPARγ in complex with different ligands. However, GW9662 exhibited an RMSD value of 2.1 Å, indicating a moderate level of structural variation in the complex. This suggests that the binding of GW9662 to PPARγ may involve some degree of flexibility. On the other hand, curcumin demonstrated an RMSD value of 1.9 Å, indicating a relatively stable binding interaction. This implies a strong and consistent interaction between curcumin and PPARγ. Interestingly, the ligands 1A6, 1A8, and 1B8 displayed RMSD values of 2.0 Å, 1.5 Å, and 2.1 Å, respectively. These values indicate varying degrees of stability in the interactions, with 1A8 showing the most stable binding.

#### 2.5.2. Assessing Structural Stability of PPARγ Protein in Its Interaction with Curcumin and Synthetic Curcumin Derivatives Using RMSF Analysis

The calculated RMSF values were used to observe the changes in protein structural flexibility as the selected compounds (GW9662, curcumin, 1A6, 1A8, and 1B8) bound to specific regions on the target (Figure 4B). Interestingly, all the selected compounds exhibited peak fluctuations in the protein at the residue positions Gly199, Ser200, His201, Met202, Gly203, and Ser204 during the simulation. These positions showed the most significant fluctuation during the binding of the compounds to the protein. Additionally, minor fluctuations were observed at Gly243, Lys244, Lys269, His270, Ile271, Thr272, Leu428, Val450, Gln452, and Tyr477. The RMSF analysis revealed that residues between Gln273 and Glu427 of the complex were the most rigid, with minimal fluctuations observed in this region.

#### 2.5.3. Assessing Structural Stability of COX2 Protein in Its Interaction with Curcumin and Synthetic Curcumin Derivatives Using RSMD Analysis

This study evaluated the conformational changes in COX2 in complex with Naproxen, curcumin, 1A6, 1A8, and 1B8. The corresponding RMSD values were observed (Figure 5A). In the absence of a ligand (APO), COX2 exhibited an RMSD value of 1.8 Å. This baseline value provides insight into the inherent stability of the protein during simulations and serves as a reference for assessing ligand-induced structural changes. Meanwhile, Naproxen exhibited an RMSD value of 2.3 Å, indicating a moderate degree of structural variation during the simulation. This suggests that the binding of Naproxen to COX2 might involve some level of flexibility as shown in Figure 5A. Likewise, curcumin displayed an RMSD value of 1.9 Å, suggesting a relatively stable binding interaction. This observation implies a robust and consistent binding between curcumin and COX2. Among the synthetic compounds, 1A6, 1A8, and 1B8, the RMSD values of 1.6 Å, 1.5 Å, and 1.8 Å were respectively observed. These values suggest varying degrees of binding stability, with 1A8 demonstrating the most stable binding interaction among the ligands.

#### 2.5.4. Assessing Structural Stability of COX2 Protein in Its Interaction with Curcumin and Synthetic Curcumin Derivatives Using RMSF Analysis

The calculated RMSF values were utilized to elucidate alterations in the structural flexibility of the protein upon interaction with the specific compounds (Naproxen, curcumin, 1A6, 1A8, and 1B8) (Figure 5B). Intriguingly, all selected compounds displayed prominent peak fluctuations within distinct regions of the protein, particularly at the residue positions Lys79, Leu80, Pro218, Ile279, Glu322, Gln370, Gln543, Val554, and Gln583 throughout the simulation period. These positions experienced the most pronounced variations upon compound binding, indicating their involvement in dynamic interactions. Furthermore, modest fluctuations were discerned in residues spanning between Leu80 and Pro218. A comprehensive RMSF analysis unveiled that residues situated between Gln370 and Gln543 of the complex exhibited elevated rigidity, with minimal fluctuations observed in this specific region.

#### 2.5.5. Assessing Structural Stability of FAS Protein in Its Interaction with Curcumin and Synthetic Curcumin Derivatives Using RMSD Analysis

This investigation delved into the conformational dynamics of FAS in complex with the ligands Orlistat, curcumin, 1A6, 1A8, and 1B8. The resulting RMSD values were meticulously analyzed as illustrated in Figure 6A. Briefly, a significant point of reference was the FAS protein in the absence of a ligand, referred to as “APO”. The APO complex exhibited an RMSD value of 2.0 Å, indicating inherent stability within the protein structure during the simulation. This value serves as a foundational measure for evaluating the dynamic behavior of FAS in complex with the ligands. Contrasting this baseline, the FAS–Orlistat complex displayed an RMSD value of 2.1 Å, suggesting moderate structural variation within the complex. This hints at a certain level of flexibility in the interaction between Orlistat and FAS. Similarly, the RMSD value for the FAS–curcumin complex was 2.1Å, indicating consistent and stable binding dynamics. Interestingly, the ligands 1A6, 1A8, and 1B8 exhibited RMSD values of 2.1 Å, 2.0 Å, and 2.1 Å, respectively. These values offer insights into varying degrees of stability within these interactions.

#### 2.5.6. Assessing Structural Stability of FAS Protein in Its Interaction with Curcumin and Synthetic Curcumin Derivatives Using RSMF Analysis

The calculated RMSF values were employed to investigate shifts in the structural flexibility of the protein upon interaction with specific compounds (Orlistat, curcumin, 1A6, 1A8, and 1B8) (Figure 6B). Remarkably, all compounds exhibited notable peak fluctuations within distinct protein regions, notably at the residues Asn2218, Val2237, Gln2238, Tyr2462, Asn2463, Lys2354, Leu2355, Thr2356, Pro2357, Gly2358, Cys2359, Gly2470, Lys2471, Val2472, Ser2473, and Val2474 throughout the simulation period. Importantly, the amino acid residues located at 2450 and 2456 of the protein structure were unidentified. This might have resulted from poor construction of the FAS protein crystal structure from X-ray crystallography. These positions demonstrated the most pronounced variations upon compound binding, underscoring their active participation in dynamic interactions. Furthermore, modest fluctuations were observed in residues spanning from Cys2359 to Gly2470. A comprehensive analysis of RMSF unveiled that residues located between Gln2238 and Tyr2462 of the complex exhibited heightened rigidity, with minimal fluctuations within this specific region.

#### 2.5.7. Calculation of the Binding Free Energy Using the MM/GBSA Approach

The binding free energies of the respective reference controls, curcumin, and the synthetic curcumin derivatives were calculated using the MM/GBSA method to measure the interaction strengths (Table 4). As indicated in Table 4, the highest binding energy was observed when curcumin interacted with PPARγ (−60.2 ± 0.4 kcal/mol) compared to other compounds for this protein. When assessing binding free energies in the COX2 systems, higher binding energy (−64.9 ± 0.2 kcal/mol) was obtained by 1A8. Interestingly, Orlistat (37.9 ± 0.3 kcal/mol) and curcumin (−37.7 ± 0.3 kcal/mol) showed similar binding free energies to FAS, with minor differences.

#### 2.5.8. Analyzing Molecular Interactions in MD Simulation Systems Using LigPlot

Based on the binding free energy calculations, an analysis of the ligand interaction plot (Table 5) was conducted to validate the intermolecular interactions responsible for the system stabilities of compounds within the binding site of human proteins involved in adipogenesis (PPARγ), inflammation (COX2), and lipid metabolism (FAS) at the end of the MD simulations. According to the interaction plot, the molecular interactions of the compounds within the binding site of PPARγ exhibit conserved interactions with PPARγ residue such as Ser88 (Table 5). Notably, GW9662 displayed one hydrogen bond interaction with Cys84 (3.22 Å) within the binding site. However, the stability of GW9662 within the binding site was primarily maintained by eight hydrophobic interactions. Like GW9662, LigPlot analysis revealed that the hydroxyl groups of curcumin established one hydrogen bond with Glu94 (2.57 Å). Additionally, seventeen amino acid residues participated in hydrophobic interactions with curcumin, thereby enhancing the stability of the system (Table 5). It is worth mentioning that 1A6 and 1B8 did not engage in hydrogen bond interactions, but their stability relied on sixteen and fifteen hydrophobic interactions within the binding site, respectively. Interestingly, the hydroxyl groups of 1A8 formed three hydrogen bond interactions with Glu94 (2.57 Å), Ser141 (3.08 Å), and Ser88 (2.83 Å) in the binding site. Furthermore, twelve hydrophobic interactions with binding site residues contributed to the system stability of 1A8 (Table 5).

The interaction plot within the binding site of the COX2 protein provided insights into the interactions of compounds, with residues Ser321, Ser498, Arg89, and Tyr323 forming significant conserved interactions with all compounds (Table 5). These interactions suggest that the compounds interacted with COX2 in a manner similar to Naproxen. It is interesting to note that all compounds formed hydrogen interactions with an Arg89 residue. Apart from two hydrogen bond interactions formed between the hydroxyl groups of Naproxen and the COX2 residue Arg89 (2.94 Å and 2.98 Å), its interactions were supported by ten hydrophobic interactions within the binding site. As previously mentioned, the hydroxyl group of curcumin engaged in hydrogen bond interactions with the COX2 residue Arg89 (3.00 Å), while relying on thirteen hydrophobic interactions to maintain stability (Table 5). Like curcumin, the hydroxyl group of 1A6 formed hydrogen bond interactions with the COX2 residue Arg89 (3.18 Å), while relying on seventeen hydrophobic interactions to maintain stability (Table 5). Exclusively, the hydroxyl groups of 1A8 formed three hydrogen interactions with the Arg89 (3.05 Å), Ser498 (2.80 Å), and Tyr353 (2.88 Å) binding site residues of COX2. In addition, its molecular interactions were further stabilized with nine hydrophobic interactions (Table 5). Like curcumin and 1A6, 1B8 formed a hydrogen bond interaction with Arg89 (2.970 Å). Additionally, twelve binding site residues contributed to the interaction stability of 1B8 through hydrophobic interactions (Table 5).

The compound interaction analysis revealed that the compounds did not form conserved interactions with the binding site residues of FAS. Interestingly, the analysis showed that Orlistat formed two hydrogen bond interactions with Arg211 (20.71 and 3.33 Å) in the binding site. However, the bonding stability of Orlistat in the FAS binding site was supported by hydrophobic interactions with six residues (Table 5). Another significant observation from the interaction analysis was the formation of two hydrogen bonds by curcumin with the Tyr130 (2.71 Å) and Glu214 (2.59 Å) residues of the binding site. In addition, the interactions of curcumin were stabilized by ten hydrophobic interactions with the binding site residues. It is interesting to note that 1A6 and 1B8 did not participate in hydrogen bonding with residues of the binding site of FAS protein. However, the stability of the interactions of 1A6 and 1B8 with the FAS protein was sustained by hydrophobic interactions with five and ten residues, respectively. Like curcumin, the hydroxyl groups of 1A8 formed hydrogen interactions with two binding site residues of FAS protein such as Tyr90 (2.86 Å) and Asp121 (2.58 Å). In addition, this stability was further maintained by hydrophobic interactions with ten binding site residues.

## 3. Discussion

Natural compounds continue to play a crucial role in drug discovery due to their vast chemical diversity and potential therapeutic value. Additionally, natural products such as plant extracts have been a significant source of bioactive compounds throughout history [24,25]. Many drugs currently on the market have their roots in natural products, including antibiotics, anticancer agents, and immunosuppressants. Their biological activity makes them valuable starting points for the development of new drugs [26,27,28,29]. Curcumin, a bioactive compound derived from the rhizome of *Curcuma longa*, has reportedly shown great potential as an anti-obesity therapeutic [30,31,32] due to its ability to modulate various cellular pathways involved in obesity including adipogenesis [12,13,14], lipid metabolism [10,11], and inflammation [15,16]. In this study, the pharmacological profiles of curcumin and its synthetic derivatives (1A6, 1A8, and 1B8) were explored to unveil insights into their potential anti-obesity activity using molecular recognition techniques.

Firstly, the characterization of curcumin and the synthetic curcumin derivatives was carried out to evaluate the pharmacokinetic, physicochemical, and toxicity properties using online predictive databases. We centered our pharmacokinetic analysis on the LRo5, which serves as a widely recognized standard for evaluating the druglikeness of chemical compounds [33]. This framework, established by Owens and Lipinski in 2003 [34] and further discussed by Pollastri et al. (2010) [35], is fundamental in characterizing potential drug candidates and has become a cornerstone in the field of drug discovery [34,35]. Our findings revealed that curcumin and its synthetic derivatives align with the LRo5 criteria, indicating their potential as drug-like substances without any violations. Lipophilicity, as an important physicochemical property of drugs and bioactive compounds, measures the affinity of a compound for lipid environments, which can affect its ability to permeate biological membranes and reach target sites. A higher lipophilicity value suggests a greater affinity for lipid-rich environments [36]. Curcumin had a lipophilicity value of 3.2, which clearly indicates that curcumin falls within the moderately lipophilic range [37,38]. Interestingly, this reporting is in line with the experimental data previously reported by Clariano et al. (2023) [39] and Shereen et al. (2019) [40], who evaluated the mechanistic delivery of curcumin in humans and estimated the lipophilicity of curcumin to be in a similar range (~LogP 3.6). The synthetic derivatives 1A6 and 1B8 demonstrated a higher lipophilicity than curcumin, potentially indicating improved membrane permeability which is advantageous for drug development to enhance bioavailability. In contrast, 1A8, notably with the lowest lipophilicity, might face challenges crossing membrane barriers effectively.

Moreover, these compounds are synthetic in nature and exhibit a favorable toxicity profile. It is worth noting that while curcumin along with 1A6 and 1B8 displayed activity towards toxicity endpoints such as hepatotoxicity and immunotoxicity at lethal dosages, their overall toxicity properties were promising. This suggests that further refinement of these compounds may mitigate these specific endpoint effects. In fact, the approach of compound refinement, aimed at reducing toxicity while enhancing effectiveness, is common in several studies [24,41,42]. The synthetic derivative 1A8 was predicted to be the safest since it did not exhibit any toxicity towards the endpoints evaluated. The observed variation in the toxicity and pharmacokinetics outcomes among the compounds could be attributed to structural modifications such as the methoxy, chloro, and phenolic functional groups on the curcumin scaffold, which explains the importance of understanding how a compound’s structure influences its biological effects [43].

Examining the metabolism and elimination patterns in the liver is a customary practice in the field of drug discovery and development. This method is crucial for anticipating the amount of a drug that effectively reaches its biological targets following oral administration, commonly referred to as the liver first-pass effect [44]. In our study, we evaluated the potential of curcumin and its synthetic derivatives to be substrates of the enzymes involved in the liver first-pass effect. The findings revealed the activity of these compounds to be common against the CYP3A4 and CYP2C9 enzymes. Of note, these enzymes are part of the cytochrome P450 family and play a crucial role in the biotransformation of various substances, including drugs in the liver [45]. Consequently, our findings suggest that curcumin and its synthetic derivatives may undergo metabolism in the liver, potentially influencing their pharmacokinetics and interaction with other drugs. These findings, especially for curcumin, greatly correlate with those previously articulated in the literature [46] and therefore open avenues for further structural modifications to steer these compounds away from becoming substrates for these biotransformation enzymes. For instance, the modification of curcumin structure with hydroxylation, methoxylation, cyclization, fluorination, or alkylation can be implemented to enhance its properties for improved stability, solubility, and biological activity.

In general, the enhanced permeability of orally administered drugs in the intestines tends to correlate with improved pharmacokinetic properties [47]. Our study illustrated that curcumin and its synthetic derivatives exhibit high absorption in the intestines, demonstrating their potential to pass into the circulatory system. The literature also indicates that curcumin metabolites such as dihydrocurcumin and tetrahydrocurcumin are traced in the intestines [48]. However, our in silico prediction for the intestinal absorption of curcumin does not correlate with previously reported in vitro and in vivo findings [49]. This may be attributed to the intestinal efflux of the P-gp enzyme on curcumin [49]. The P-gp is a vital membrane protein in the liver, intestines, and other tissues that expels substances from cells [50]. Our study predicted that curcumin and its synthetic derivatives are not substrates for the P-gp enzyme, indicating that this enzyme does not play a role in expelling these compounds from cells. Unfortunately, these predictive findings, mainly on curcumin, are opposed by experimental data in the literature [49]. In fact, a recent in silico study suggested that synthetic curcumin derivatives interact with the P-gp enzyme as antagonist [51].

The selection of protein targets in this study was vital for investigating the mechanism of action of curcumin and its synthetic derivatives and predicting their potential as anti-obesity therapeutic agents. The systematic approach used predicted targets that are shared by all compounds and are relevant to the metabolic pathways involved in obesity including adipogenesis, lipid metabolism, and inflammation. Based on molecular docking results (binding interaction and docking score), three target proteins, PPARγ, COX2, and FAS, were best docked to curcumin and the synthetic derivatives among the nine target proteins that were selected and evaluated for molecular docking. Moreover, the results showed that the binding interactions and docking affinity between curcumin and the synthetic derivatives were similar for PPARγ, COX2, and FAS. This similarity in binding modes indicates that these derivatives share a common mechanism of action with curcumin, possibly due to structural similarity. Additionally, these compounds exhibited comparable binding affinity to PPARγ, COX2, and FAS when compared to the respective reference ligands (GW9962, Orlistat, and Naproxen).

In the PPARγ complex, curcumin and its synthetic derivatives maintained favorable binding to the target protein. In contrast, GW9662, which served as a reference control, exhibited the lowest binding affinity to PPARγ when compared to curcumin and its synthetic derivatives. This suggests that curcumin and its synthetic derivatives may be potential ligands for PPARγ. PPARγ is a nuclear receptor transcription factor that plays a key role in adipogenesis [52] and is a major therapeutic target for various diseases including obesity, dyslipidemia, type 2 diabetes mellitus, neurodegenerative disorders, and cancers [53]. In experimental studies, curcumin and synthetic analogs of curcumin were demonstrated to induce PPARγ expression in glial and carcinoma cells, thereby providing neuroprotective and anti-cancer effects [54,55]. This further suggests that other than being potential ligands, curcumin and its derivatives may regulate the expression of PPARγ.

In the COX2 complex, 1B8 demonstrated a high docking score compared to Naproxen, while curcumin indicated a similar binding affinity to Naproxen. However, 1A6 and 1A8 demonstrated the lowest docking scores. This result suggests that curcumin and its derivative 1B8 may have a higher binding affinity for COX2 than Naproxen. Compared to our study, Sohilait et al. (2017) [56] showed that analogs of curcumin strongly bind with COX2. Although 1B8 had the most favorable affinity towards COX2 compared to curcumin and other synthetic derivatives, these findings suggest that 1B8 can be a potential ligand for COX2 regulation.

Orlistat, used as the reference control, exhibited the weakest binding affinity for the FAS protein when compared to curcumin and its derivatives. FAS is an enzyme involved in the synthesis of fatty acids by catalyzing the synthesis of long-chain fatty acids from acetyl-CoA and malonyl-CoA [57]. HoweverMoreover, Orlistat, an anti-obesity agent that inhibits gastric and pancreatic lipases, has been experimentally shown to inhibit FAS expression [58]. This implies that Orlistat is particularly effective in binding to FAS, making it a potent inhibitor of this enzyme as recently demonstrated [59]. An in vivo study by Maithilikarpagaselvi et al. (2016) [60] demonstrated that curcumin facilitates its anti-obesity effects by suppressing fat accumulation through the inhibition of FAS enzymes.

Overall, the difference in binding affinity between curcumin and its derivatives was significantly minimal, which could be due to the similarity in the structural scaffold. This structural similarity likely results in comparable binding interactions with the target protein. The observation of similar binding affinities among curcumin and its derivatives can have important implications for drug development. It suggests that while structural modifications might lead to improvements in other aspects (e.g., pharmacokinetics, toxicity, or selectivity), they may not have a pronounced effect on the binding affinity.

To elucidate on the stability of these interactions, molecular dynamics simulations were conducted for both unbound and bound forms of PPARγ, COX2, and FAS. The results revealed that all simulated systems remained stable throughout the 200 nanoseconds of the simulation. Stability was confirmed by minimal atomic deviations, which were within the generally accepted threshold range of 1 to 3 angstroms for good stability [61]. This stability implies that the interactions between the compounds and their respective protein targets were robust and maintained throughout the simulation, providing confidence in the reliability of the binding interactions. In the specific case of PPARγ, the analysis indicated that curcumin, synthetic curcumin derivatives, and GW9662 induced structural flexibility in similar regions of the target protein. This flexibility was characterized by increased fluctuations in these regions when compared to the unbound state of the target. Such fluctuations suggest that the binding of these compounds caused PPARγ to adopt a more flexible conformation, which may have functional implications related to its mode of action. Generally, conformational dynamics of PPARγ inhibitors can be influenced by factors such as ligand structure, binding site interactions, and solvent environment [62,63]. Different conformations of inhibitors can have varying degrees of affinity and specificity towards the PPARγ protein [64].

In the COX2 and FAS systems, curcumin and the synthetic curcumin derivatives showed similar conformational dynamics on each of the targets, which was characterized by increased structural flexibility and a less compact structure in contrast to the unbound targets. The findings could therefore suggest the protein is undergoing more dynamic conformational changes, allowing it to explore a wider range of structural configurations [65,66]. Similarly, FAS and COX2 inhibitors exhibited varying conformational changes to their respective targets depending on the structures of their respective inhibitors [67,68]. Overall, curcumin and the synthetic curcumin derivatives had a similar impact on the conformational dynamics of the target proteins, which is attributed to the similarity in their structures.

In addition, we assessed the amino acids within the binding site responsible for stabilizing molecular interactions following molecular dynamics simulations. This analysis is crucial in drug discovery to tailor molecules that selectively interact with the target protein [69]. Our results revealed that compounds interacting with the binding site of PPARγ consistently engage in specific interactions with residues like Ser88, affirming their association with the localized region of the PPARγ binding site. Notably, the binding of curcumin was sustained through hydrophilic interactions with seventeen amino acid residues and one hydrogen interaction in the binding site, potentially contributing to the observed high binding free energies. The interaction between curcumin and PPARγ has been well-documented in various studies [70,71,72]. Furthermore, synthetic derivatives of curcumin exhibited stronger binding free energies compared to the reference compound (GW9662), supported by their hydrophilic and hydrogen interactions and amino acid residues within the binding site. This is speculated to result from the structural similarity shared by curcumin and its derivatives. One study suggested that curcumin analogs interact with PPARγ to suppress adipogenesis more effectively than curcumin [73]. This implies that derivatives of curcumin may outperform curcumin in targeting the PPARγ isoform regulating adipogenesis, while activating the PPARγ isoform associated with inflammation and lipid metabolism suppression. The observed variations underscore the complexity of the molecular interactions and highlight the potential therapeutic implications of curcumin derivatives in modulating PPARγ isoforms.

In this study, we demonstrated consistent interaction of the ligand compounds with amino acid residues like Ser321, Ser498, Arg89, and Tyr323, forming conserved interactions within the binding site of COX2. Remarkably, the synthetic derivative 1A8 exhibited the highest binding free energy, greater than the reference compound (Naproxen). This strong interaction was substantiated by nine hydrophilic and three hydrogen interactions with amino acid residues in the COX2 binding site. The tendency of curcumin analogs to strongly interact with the COX2 binding site has been previously reported [56], while curcumin derivatives have also been reported to selectively interact with COX2 [74]. This specificity may account for the superior interaction demonstrated by 1A8 compared to 1A6 and 1B8. Despite 1A8 outperforming other compounds, it is essential to recognize that curcumin exhibited strong interactions with COX2, surpassing the control inhibitor (Naproxen). Compared to this study, an in vivo study demonstrated that curcumin inhibits the expression of COX2 [75,76].

While common interactions were observed for PPARγ and COX2, the ligand compounds did not engage with shared amino acid residues in the binding site of FAS. However, curcumin demonstrated two hydrogen interactions with Tyr130 (2.71 Å) and Glu214 (2.59 Å) residues within the binding site residue of FAS. Additionally, the interaction of curcumin within the FAS binding site was stabilized by ten hydrophilic interactions. This interaction was validated by high binding free energies, comparable to the reference compound (Orlistat). Interestingly, an in vitro study on the breast cancer cell line SKBR3 also demonstrated that curcumin decreases the activity and expression of the FAS enzyme [77]. It is crucial to highlight that curcumin displayed substantial interactions with both PPARγ and FAS, possibly due to the involvement of these proteins in lipid metabolism. Curcumin derivatives have been shown to decrease FAS expression, thus suppressing lipid accumulation in HepG2 cells [78]. This indicates that curcumin and its synthetic derivatives potentially target FAS and regulate its expression.

While our study presents promising findings in the comprehensive analysis of curcumin and its synthetic derivatives, several limitations should be acknowledged. The reliance on computational analyses, without direct in vitro and in vivo validation, leaves a gap in understanding the clinical implications of our results. Furthermore, the scope of toxicity assessment is primarily focused on acute effects, necessitating extended studies to capture potential long-term impacts and interactions with other physiological systems. In addition, caution is warranted in generalizing the findings, as the study centers on specific derivatives (1A6, 1A8, and 1B8), while variability among curcumin derivatives may exist. Despite overall promising results, specific endpoint effects observed at lethal dosages highlight the need for refinement strategies to enhance the safety profile of the compounds. Additionally, the focus of the study on a select few anti-obesity target proteins may limit the generalizability of the results, and further exploration of interactions with a broader range of relevant proteins is recommended. Lastly, while the study identifies potential interactions with key liver enzymes, a more detailed exploration of the metabolic pathways undergone by these compounds is necessary.

This in-depth investigation combines computational methodologies with molecular recognition techniques to unravel the potential anti-obesity activity of curcumin and its synthetic derivatives. The findings provide a foundation for further experimental validation, emphasizing the need for a holistic understanding of the pharmacological profiles of these compounds for effective drug development. The structural insights and mechanistic understanding garnered from this study pave the way for future investigations bringing research close to harnessing the therapeutic potential of curcumin and its derivatives in the fight against obesity.

## 4. Materials and Methods

### 4.1. Prediction of Toxicological Properties of Curcumin and Synthetic Curcumin Derivatives

ProTox II is a free webserver that is used to computationally predict the toxicity of compounds in a biological system [79]. The canonical simplified molecular-input line-entry system (SMILES) of curcumin and three synthetic curcumin derivatives (1A6, 1A8, and 1B8) were submitted to the ProTox II webserver, as shown in Figure 7, to define their toxicological endpoints. These include hepatotoxicity, carcinogenicity, immunotoxicity, and cytotoxicity. ProTox II was also used to predict the median lethal doses (LD_50_) values using the globally harmonized system of classification of labelling of chemicals (GMS) [79,80]. Furthermore, the predicted toxicity classifications were also assessed. These include extremely toxic (low LD_50_), highly toxic (moderate LD_50_), moderately toxic (intermediate LD_50_), slightly toxic (high LD_50_), and practically non-toxic (very high LD_50_) [81].

### 4.2. Predicting the Pharmacokinetic, Physicochemical, and Druglikeness Properties of Curcumin and Synthetic Curcumin Derivatives

SwissADME [82], which is a free webserver used to predict the pharmacokinetics, druglikeness, and medicinal chemistry properties of a compound, was used to determine the general chemical parameters of curcumin and the three synthetic curcumin derivatives (1A6, 1A8, and 1B8) as indicated in Figure 7. The physicochemical properties such as lipophilicity (iLOGP) and water solubility (ESOL) were defined, while the gastrointestinal (GI) absorption, which is a pharmacokinetic parameter, was also predicted. The effect of curcumin, 1A6, 1A8, and 1B8 on the liver drug efflux pump (otherwise known as P-glycoprotein (P-gp) substrate) and the inhibition of cytochrome P450 enzymes (CYP3A4, CYP1A2, CYP2D6, and CYP2C9), which are involved in liver metabolism was assessed [83]. Drug parameters such as druglikeness using Lipinski’s rules (LRo5) [84] and synthetic score were also defined.

### 4.3. Identification of the Potential Biological Targets of Curcumin and Synthetic Curcumin Derivatives

To determine the potential biological targets of curcumin, 1A6, 1A8, and 1B8, the canonical SMILES of these compounds were submitted to SwissTarget Prediction [85] and SEA (similarity ensemble approach) [86] as shown in Figure 7, which are free webservers that identify possible target proteins based on the published experimental literature of the compound or related compounds. In addition, the online databases PubMed and Google Scholar were used to search for studies reporting on the proteins that are targeted by curcumin and regulate adipogenesis, lipid metabolism, and inflammation in adipose tissue. In this regard, three common potential biological targets of these compounds or those that are at least targeted by curcumin in each pathway were selected and used for downstream experiments. The selected target proteins were then used for molecular docking studies to assess the interaction and binding affinities of the compounds (curcumin, 1A6, 1A8, and 1B8) with these target proteins.

### 4.4. Molecular Docking of Curcumin and Synthetic Curcumin Derivatives to the Target

#### 4.4.1. Generation of the Three–Dimensional Structure and Energy Minimization of the Compounds

The two-dimensional structures of curcumin and the three synthetic curcumin derivatives were drawn, whereafter their three-dimensional structures were generated and energy minimized using the ACD/ChemSketch freeware [87]. Experimentally established ligands (PF05221304, Rosiglitazone, Raloxifene, and LY2228820) or Food and Drug Administration (FDA)-approved drugs (BBI608, GW9662, Orlistat, Naproxen, and FT30356) of corresponding target proteins were used as references to confirm the docking poses of the compounds. All generated three-dimensional structures were saved in .mol2 format. Thereafter, the three-dimensional structures were submitted to Avogadro version 2.0 software for optimization [88]. The Avogadro software is used to optimize the three-dimensional structures of compounds by adding partial charges using Ghemical and Merck molecular force fields 94 (MMFF94s) and then afterwards minimizing energy to yield a more optimal geometric structure of compounds. All optimized compounds were saved in .mol2 format in preparation for molecular docking.

#### 4.4.2. Protein Preparation

The X-ray crystal structures of the target proteins (Table 6) were identified online using the RSCB Protein Data Bank (PDB) [89] and downloaded in .pdb format. The UCSF Chimera Software package, version 1.17.3 was used to optimize the protein structures for molecular docking by removing water molecules and all other non-standard residues [90]. Thereafter, the optimized protein structures were saved in PBD format in preparation for molecular docking.

#### 4.4.3. Identification of the Binding Sites of the Predicted Protein Targets

Crystal structures of target proteins retrieved from the PDB with unknown ligand-binding sites were submitted to the Computed Atlas of Surface Topography of proteins (CASTp) free webserver, using their PDB ID (5F1A, 1ALU, and 6U66), for the prediction of binding sites. CASTp is an online protein analytic tool that assesses the surface topography of a protein. The CASTp tool also identifies and highlights pockets and cavities in the three-dimensional crystal structure of a protein [91]. Pockets predicted to have a high surface area and volume were considered as potential interactive sites responsible for the binding of ligands.

#### 4.4.4. Molecular Docking Studies

The purpose of molecular docking is to assess the binding affinity of two molecules using various scoring algorithms [92]. The Autodock Vina Plugin found on UCSF Chimera as indicated in Figure 7 was used to dock the generated three-dimensional structures to the target proteins, thus defining the best ligand–protein binding conformation as the output. For each target protein, molecular docking simulations were carried out with the docking grid box containing different co-ordinates as shown in Table 7. After molecular docking, the docked complexes were arranged in consistent order, while reference compounds were outlined first, and the root mean square from the original docking mode of each compound was determined. Superimposition was carried out to confirm that compounds were docked within the identified conserved regions.

### 4.5. Molecular Dynamic Simulations of Curcumin and Synthetic Curcumin Derivatives

The SANDER engine provided as part of the AMBER package was used for molecular dynamic (MD) simulations, and the AMBER force field, FF14SB [93], was utilized to describe the systems. To determine atomic partial charges for the ligand, we utilized ANTECHAMBER along with the restrained electrostatic potential (RESP) and general amber force field (GAFF) procedures. Hydrogen atoms were added to the systems using the Leap module of AMBER 14. Neutralization was attained by adding 6 Na^+^ on the PPARγ and FAS protein structures, and 1 Na^+^ on the COX2 protein structure. The amino acids were renumbered based on the dimeric form of the protein, resulting in residues numbered from 1 to 276 for PPARγ, 1 to 551 for COX2, and 1 to 275 for FAS. To create the simulation systems, the systems were placed within an orthorhombic box filled with TIP3P water molecules. The positioning ensured that all atoms were within 10 Å of any box edge.

An initial minimization was performed with a restraint potential of 500 kcal/mol applied to both substances of interest (solutes). The minimization consisted of 1000 steps using the steepest descent method followed by 1000 steps of conjugate gradients. An additional full minimization of 1000 steps were carried out using the conjugate gradient algorithm without any restraints. The systems were gradually heated from 0 K to 310.5 K (physiological temperature) during a 0.05 ns MD simulation. The number of atoms and volume of the systems were kept fixed. The solutes within the systems were subjected to a potential harmonic restraint of 10 kcal/mol and a collision frequency of 0.001 ns. Following the heating phase, an equilibration step of approximately 0.5 ns was performed for each system. The temperature was maintained at 310.5 K during this equilibration process. The systems were simulated under an isobaric–isothermal ensemble (NPT), with several features kept constant, including the number of atoms and pressure. The pressure of the system was maintained at 1 bar using the Berendsen barostat.

The MD simulations were conducted for 200 ns through AMBER 18 using the GPU version as indicated in Figure 7. In each simulation, the SHAKE algorithm was employed to constrain the bonds involving hydrogen atoms. The time step for the simulations was set to 2 fs, and an SPFP precision model was used. The simulations followed the isobaric–isothermal ensemble (NPT) conditions, with randomized seeding, a constant pressure of 1 bar maintained by the Berendsen barostat, a pressure-coupling constant of 0.002 ns, a physiological temperature, and a Langevin thermostat with a collision frequency of 0.001 ns.

### 4.6. Post-Molecular Dynamic Simulations Analysis

After saving the coordinates of all eighteen systems, trajectory analysis was performed at every 100 ns interval using PTRAJ. The analysis included calculating the root mean square deviation (RMSD) and root mean square fluctuation (RMSF). These analyses were conducted using the CPPTRAJ module. For molecular interaction analysis, Molegro Molecular Viewer version 2.2 [94] and LigPlot version 2.2.8 [95] software were utilized to define ligand binding interactions and intermolecular interactions within the binding site of the protein at the end of the simulation period (200 ns) as indicated in Figure 7.

### 4.7. Binding Free Energy Calculations

To evaluate and compare the binding affinity of the systems, the binding free energy was determined using the molecular mechanics/generalized born surface area method (MM/GBSA) [96], as shown in Figure 7. The calculation of the binding free energy involved averaging 200 snapshots extracted from the 200 ns trajectory. The binding free energy (ΔG) obtained through this method [97] for each molecular species (complex, ligand, and receptor) can be described as follows:(1)ΔGbind = Gcomplex − Greceptor – Gligend;(2)ΔGbind = Egas + Gsol -ST;(3)Egas = Eint + Evdw + Eele;(4)Gsol = GGB + GSA;(5)GSA = ΔSASA.

The term “Egas” represents the energy in the gas phase, which includes the internal energy “Eint”, the Coulomb energy “Eele”, and the Van der Waals energy “Evdw”. The Egas value was directly calculated using the FF14SB force field terms. The solvation free energy, “Gsol”, was estimated by considering the energy contributions from both polar states, “GGB”, and non-polar states, “G”. The non-polar solvation energy, “GSA”, was determined based on the solvent accessible surface area (SASA) using a water probe radius of 1.4 Å. On the other hand, the polar solvation contribution, “GGB”, was estimated by solving the generalized born (GB) equation. The terms “S” and “T” represent the total entropy of the solute and the temperature, respectively.

### 4.8. Data Analysis

OriginPro version 9.1 data analysis software [98] was used to generate plots for all raw data. Discovery Studio Visualizer version 21.1.0.20298 [99] and LigPlot+ version 2.2 [100] software were used for visualization.

## 5. Conclusions

In conclusion, our comprehensive analysis comparing curcumin with its synthetic derivatives reveals promising findings across pharmacokinetic, toxicity, and physicochemical parameters. The derivatives, namely 1A6, 1A8, and 1B8, align with druglikeness criteria, showcasing potential as drug candidates. Physicochemical properties, particularly lipophilicity, indicate variations that could impact membrane permeability and bioavailability, with 1A8 potentially offering improved membrane penetration. Toxicity profiles highlight overall promise, with 1A8 demonstrating the safest profile. However, refinement strategies for all compounds are recommended to mitigate specific endpoint effects observed at lethal dosages. Metabolism analyses suggest potential interactions with key liver enzymes, emphasizing the importance of structural modifications to optimize pharmacokinetics and minimize potential drug interactions. Molecular docking and molecular dynamics simulations underscore the similarities in binding affinities and interactions among curcumin and its derivatives with pivotal anti-obesity target proteins—PPARγ, COX2, and FAS. Stability throughout the simulations and identified amino acid interactions within binding sites support the robustness of these interactions.

Collectively, the in silico exploration of the pharmacokinetic and pharmacodynamic profiles of curcumin and its synthetic derivatives reveals insightful mechanisms, suggesting their potential as anti-obesity agents by targeting key metabolic pathways. Although our computational results suggest a similarity in interactive activity between curcumin and its synthetic derivatives, it is crucial to conduct additional experimental validation to enhance our understanding of their regulatory capabilities. In vitro experiments, including gene and protein expression assays, should be undertaken on the identified biological targets to explore the mechanisms of action associated with these compounds and validate the findings of this study. The structural insights and mechanistic understanding derived from this study provide valuable guidance for future research and optimization efforts aimed at harnessing the therapeutic capabilities of these compounds.

## Figures and Tables

**Figure 1 ijms-25-02603-f001:**
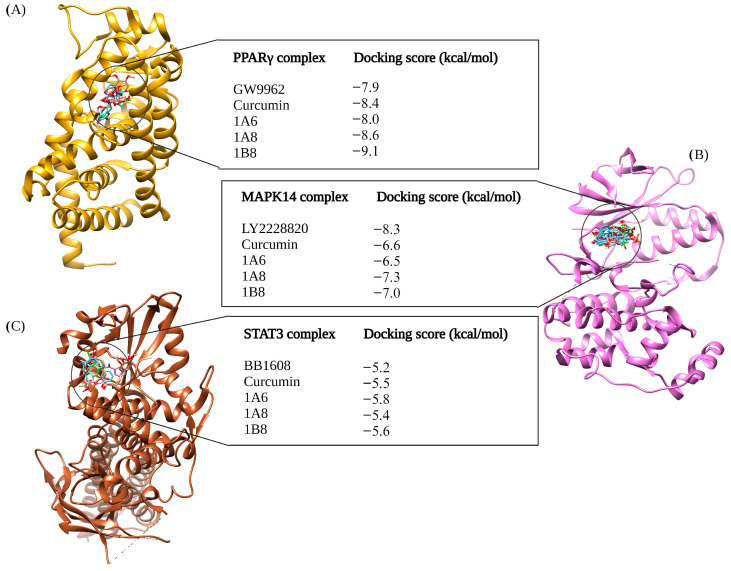
In silico binding affinity scores and superimposed docked complexes of curcumin and its synthetic curcumin derivatives (1A6, 1A8, and 1B8) bound to human target proteins involved in adipogenesis, including PPARγ (PDB ID: 7WGO), MAPK14 (PDB ID: 5XYX), and STAT3 (PDB ID: 6NUQ). All three target proteins were also docked with experimentally identified inhibiting standards; (**A**) GW9662 was docked to PPARγ with a binding energy of −7.9 kcal/mol, (**B**) LY2228820 was docked to MAPK14 with binding energy of −8.3 kcal/mol, and (**C**) BB1608 was docked to STAT3 with a binding energy of −5.2 kcal/mol. Abbreviations: MAPK14—mitogen-activated protein kinase 14; PPARγ—peroxisome proliferator-activated receptor gamma; STAT3—signal transducer and activator of transcription 3. Created with BioRender.com (accessed on 19 November 2023).

**Figure 2 ijms-25-02603-f002:**
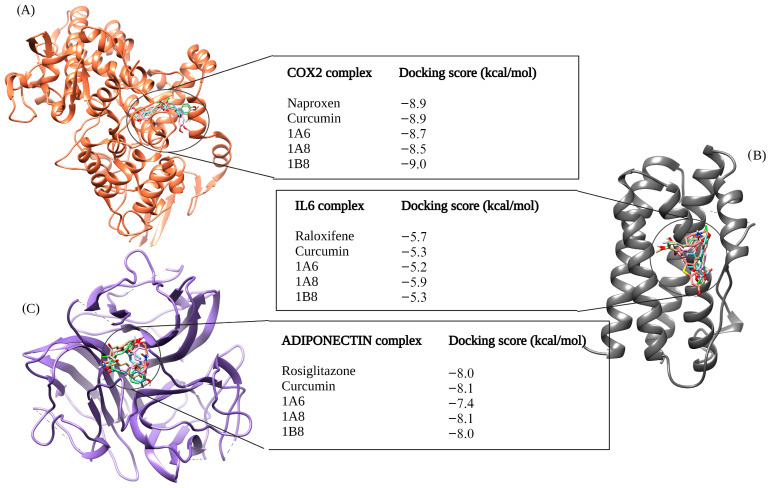
In silico binding affinity scores and superimposed docked complexes of curcumin and synthetic curcumin derivatives (1A6, 1A8, and 1B8) bound to human target proteins of inflammation such as COX2 (PDB ID: 5F1A), IL6 (PDB ID: 1ALU), and Adiponectin (PDB ID: 6U66). All three target proteins were docked with experimentally established inhibiting standards; (**A**) Naproxen was docked to COX2 with a binding energy of −8.9 kcal/mol, (**B**) Raloxifene was docked to IL6 with binding energy of −5.7 kcal/mol, and (**C**) Rosiglitazone was docked to Adiponectin with a binding energy of −8.0 kcal/mol. Abbreviations; COX2—cyclooxygenase 2, IL6—interleukin 6. Created with BioRender.com (accessed on 19 November 2023).

**Figure 3 ijms-25-02603-f003:**
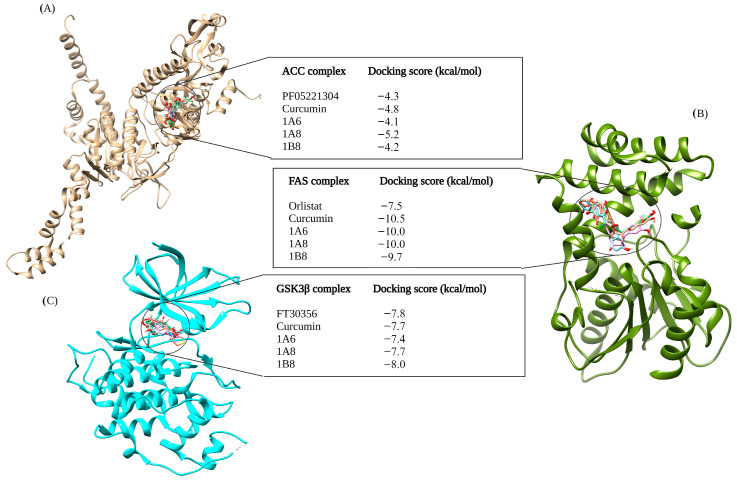
In silico binding affinity scores and superimposed docked complexes of curcumin and synthetic curcumin derivatives (1A6, 1A8, and 1B8) bound to human target proteins involved in lipid metabolism such as ACC (PDB ID: 3TDC), FAS (PDB ID: 3TJM), and GSK3β (PDB ID: 5F95). All three target proteins were docked with experimentally established inhibiting standards; (**A**) PF0522104 was docked to ACC with a binding energy of −4.3 kcal/mol, (**B**) Orlistat was docked to FAS with binding energy of −7.5 kcal/mol, and (**C**) FT30356 was docked to GSK3β with a binding energy of −7.8 kcal/mol. Abbreviations; ACC—acetyl-CoA carboxylase, FAS—fatty acid synthase, GSK3β—glycogen synthase kinase 3beta. Created with BioRender.com (accessed on 19 November 2023).

**Figure 4 ijms-25-02603-f004:**
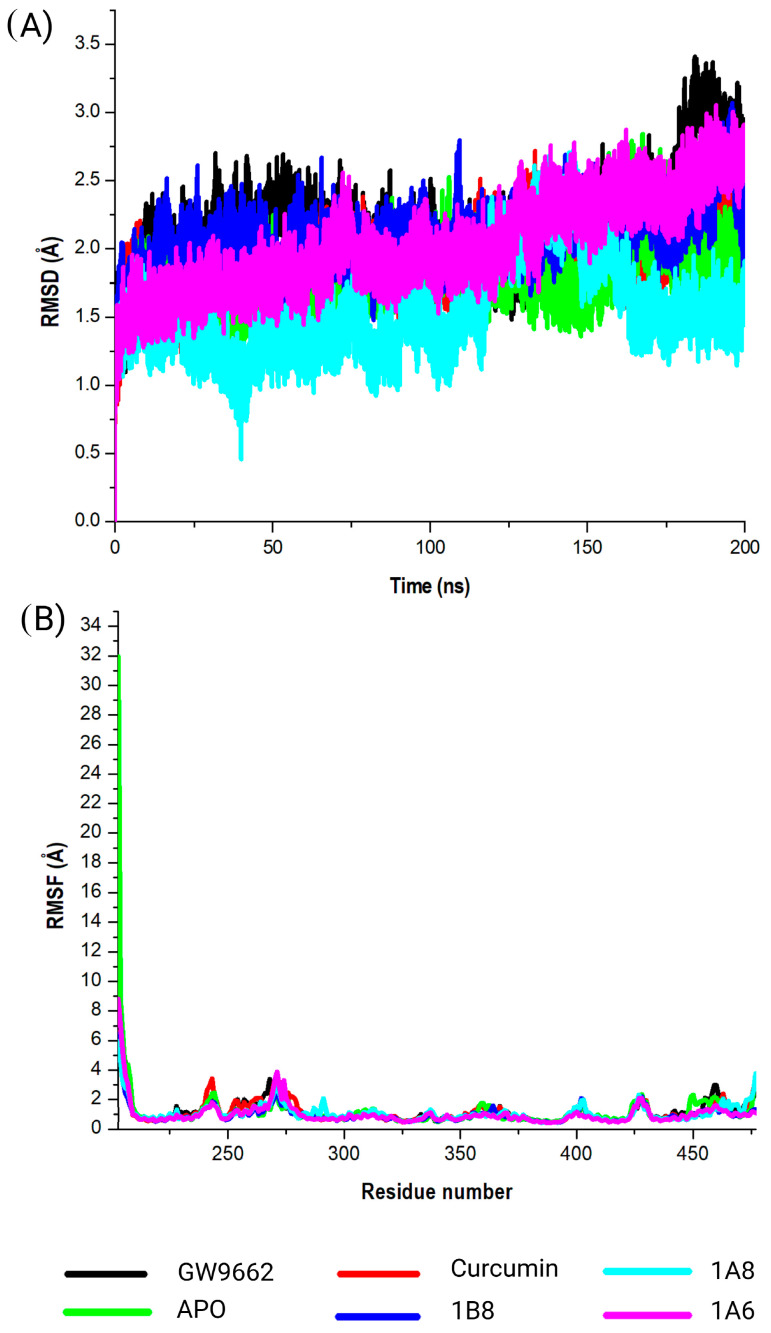
Analysis of the MD simulations of PPARγ in complex with GW9662, curcumin, 1A6, 1A8, and 1B8. (**A**) demonstrates the RMSD values of the MD simulated systems, while (**B**) illustrates the RMSF values of the MD simulated systems. The native PPARγ is referred to as APO.

**Figure 5 ijms-25-02603-f005:**
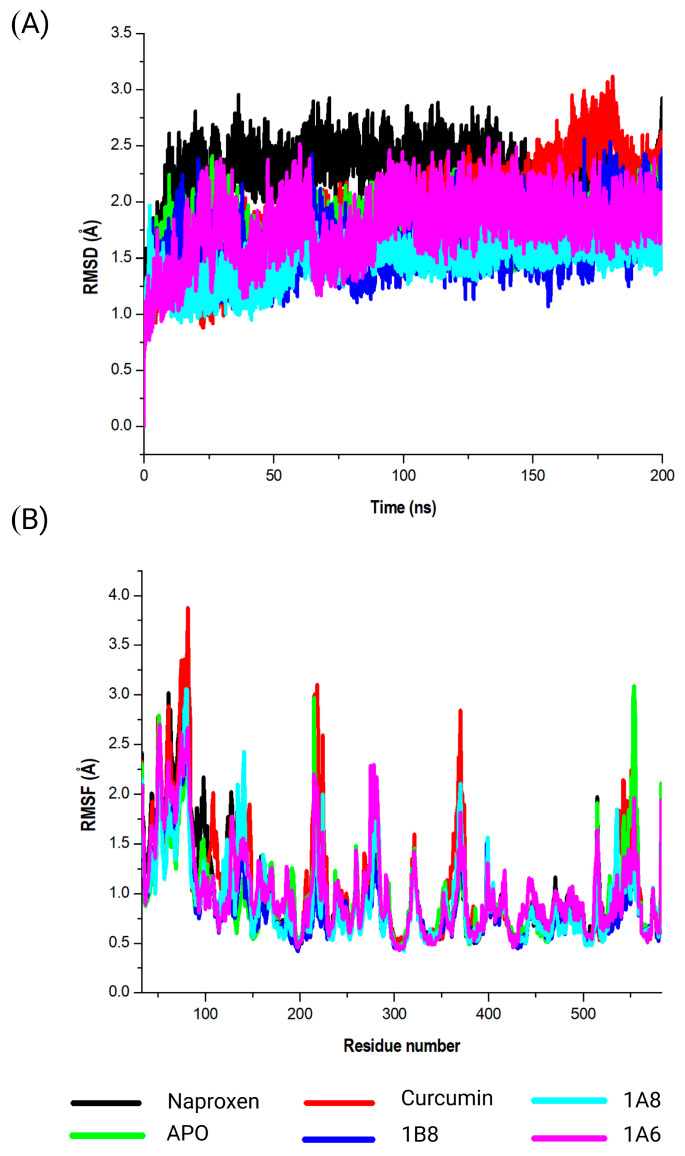
Analysis of the MD simulations of COX2 in complex with Naproxen, curcumin, 1A6, 1A8, and 1B8. (**A**) demonstrates the RMSD values of the MD simulated systems, while (**B**) illustrates the RMSF values of the MD simulated systems. The native COX2 is referred to as APO.

**Figure 6 ijms-25-02603-f006:**
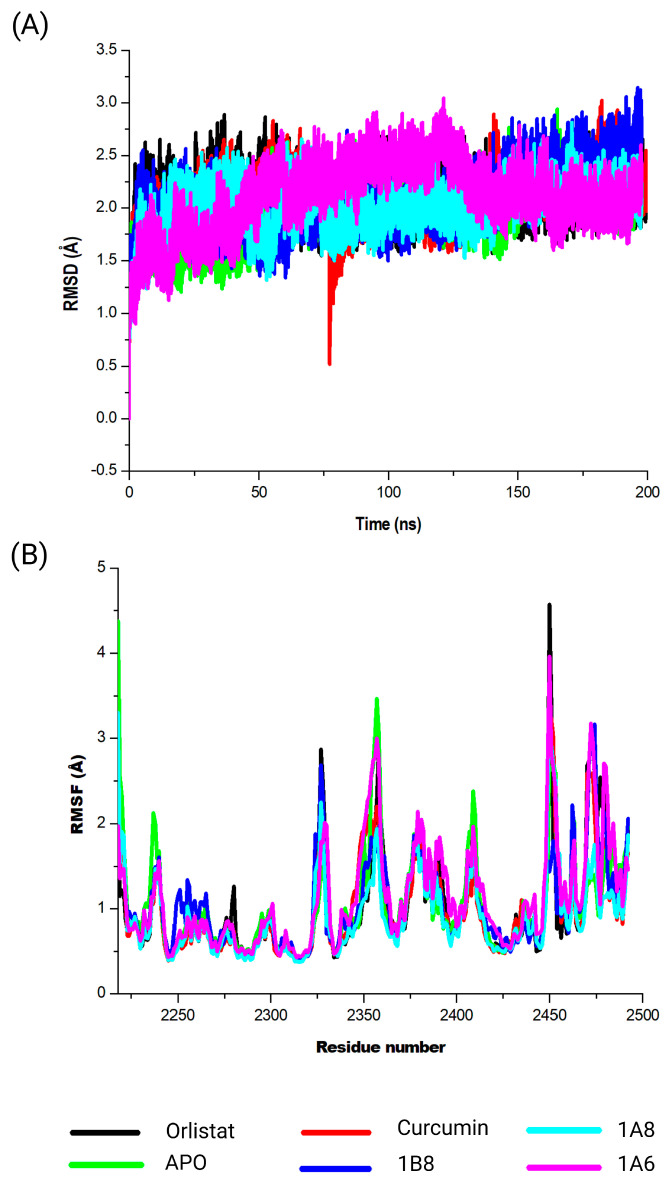
Analysis of the MD simulations of FAS in complex with Orlistat, curcumin, 1A6, 1A8, and 1B8. (**A**) demonstrates the RMSD values of the MD simulated systems, while (**B**) illustrates the RMSF values of the MD simulated systems. The native FAS is referred to as APO.

**Figure 7 ijms-25-02603-f007:**
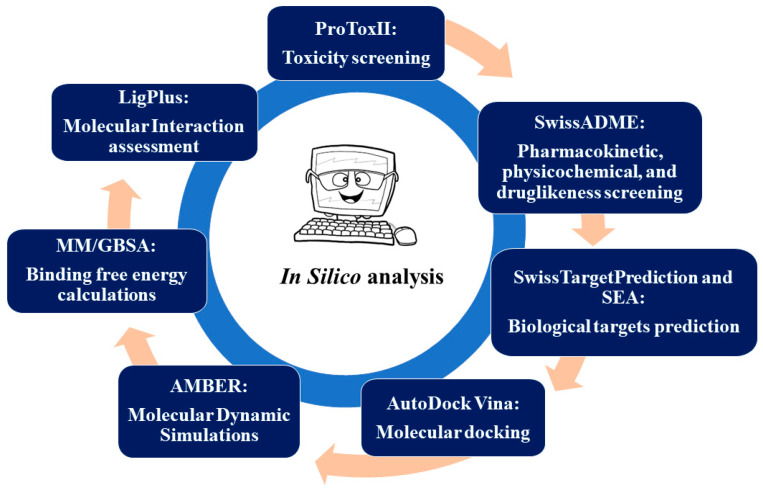
The schematic diagram of the study design.

**Table 1 ijms-25-02603-t001:** Predicted toxicity profiles of curcumin, 1A6, 1A8, and 1B8.

Toxicity	Curcumin	1A6	1A8	1B8
LD_50_ (mg/kg)	2000	4000	4400	4000
Class	4	5	5	5
Hepatotoxicity	Inactive; *p* (0.61)	Inactive; *p* (0.52)	Inactive; *p* (0.63)	Active; *p* (0.53)
Carcinogenicity	Inactive; *p* (0.84)	Inactive; *p* (0.64)	Inactive; *p* (0.60)	Inactive; *p* (0.67)
Immunotoxicity	Active; *p* (0.92)	Active; *p* (0.80)	Inactive; *p* (0.61)	Active; *p* (0.96)
Cytotoxicity	Inactive; *p* (0.88)	Inactive; *p* (0.70)	Inactive; *p* (0.75)	Active; *p* (0.73)

It should be noted that these toxicity assessments are based on lethal dose predictions. P-value refers to the probability of activity or inactivity.

**Table 2 ijms-25-02603-t002:** Predicted pharmacokinetic profiles of curcumin and 1A6, 1A8, and 1B8.

	Curcumin	1A6	1A8	1B8
Lipophilicity (iLOGP)	3.27	3.88	1.85	3.83
Water solubility (ESOL, mg/mL)	−3.94	−5.03	−4.45	−5.16
GI absorption	High	High	High	High
P-gp substrate	No	No	No	No
CYP3A4	Yes	Yes	Yes	Yes
CYP1A2	No	No	Yes	Yes
CYP2D6	No	Yes	No	No
CYP2C9	Yes	Yes	Yes	Yes

**Table 3 ijms-25-02603-t003:** The predicted chemical parameters of curcumin, 1A6, 1A8, and 1B8.

	Curcumin	1A6	1A8	1B8
Molecular weight	368.38 g/mol	430.88 g/mol	342.77 g/mol	376.44 g/mol
Number of hydrogen bond acceptors	6	6	4	4
Number of hydrogen bond donors	2	0	2	0
Octanol/water partition coefficient(logP)	3.37	4.19	3.57	4.91
Druglikeness (Lipinski)	Yes; 0 violation	Yes; 0 violation	Yes; 0 violation	Yes; 0 violation
Synthetic score	2.97	3.76	3.27	3.46

**Table 4 ijms-25-02603-t004:** The analysis of the MM/BGSA-based binding free energy contributed by residues of the active site of protein–ligand complex.

Systems		Energy Components (kcal/mol)				
		ΔE_vdw_	ΔE_ele_	ΔG_gas_	ΔG_solv_	ΔG_bind_
PPARγ	GW9662	−14.8 ± 0.4	−0.9 ± 1.6	48.0 ± 1.5	−59.0 ± 1.4	−11.0 ± 0.4
	Curcumin	−60.9 ± 0.3	−42.1 ± 0.6	−103.0 ± 0.6	42.8 ± 0.3	−60.2 ± 0.4
	1A6	−60.9 ± 0.2	−13.3 ± 0.3	−74.2 ± 0.3	23.9 ± 0.2	−50.3 ± 0.3
	1A8	−43.1 ± 0.3	−29.4 ± 0.4	−72.5 ± 0.3	30.0 ± 0.2	−42.5 ± 0.3
	1B8	−55.9 ± 0.3	−17.3 ± 0.4	−73.9 ± 0.5	26.8 ± 0.4	−46.5 ± 0.3
COX2	Naproxen	−37.1 ± 0.2	−24.3 ± 0.6	−61.3 ± 0.7	26.7 ± 0.5	−34.6 ± 0.2
	Curcumin	−48.8 ± 0.2	−24.1 ± 0.5	−72.9 ± 0.5	31.0 ± 0.4	−41.9 ± 0.3
	1A6	−57.9 ± 0.2	−15.7 ± 0.3	−73.6 ± 0.4	24.8 ± 0.2	−48.9 ± 0.3
	1A8	−18.4 ± 0.2	−216.6 ± 0.7	−142.0 ± 0.6	77.1 ± 0.5	−64.9 ± 0.2
	1B8	−55.5 ± 0.1	−20.1 ± 0.3	−75.5 ± 0.5	31.2 ± 0.3	−44.3 ± 0.3
FAS	Orlistat	−39.0 ± 0.2	−18.1 ± 0.5	−57.2 ± 0.5	19.3 ± 0.4	−37.9 ± 0.3
	Curcumin	−44.2 ± 0.3	−20.5 ± 0.6	−64.7 ± 0.6	27.0 ± 0.5	−37.7 ± 0.3
	1A6	−40.5 ± 0.6	−10.2 ± 0.4	−50.7 ± 0.7	20.4 ± 0.3	−30.3 ± 0.6
	1A8	−35.7 ± 0.2	−25.6 ± 0.6	−61.3 ± 0.6	29.8 ± 0.5	−31.5 ± 0.3
	1B8	−42.6 ± 0.3	−11.8 ± 0.3	−54.4 ± 0.5	21.4 ± 0.3	−33.0 ± 0.4

ΔEvdw—van der Waals forces; ΔEele—electrostatic interactions; ΔGbind—binding free energy; ΔGgas—gas phase interaction; ΔGsol—solvation energy.

**Table 5 ijms-25-02603-t005:** The LigPlot analysis for the molecular interactions of curcumin, 1A6, 1A8, 1B8, and their corresponding reference standards bound to PPARγ, COX2, and FAS.

Compound Name	Hydrophobic Interactions	Hydrogen Bond (Length-Å)
Adipogenesis (PPARγ)		
GW9662	Ile80, Leu129, Leu132, Ile140, Arg87, Ala91, Ser88, Ile125	Cys84 (3.22)
Curcumin	His248, Tyr126, Leu252, Tyr272, Leu264, Ser88, Cys84, Leu132, Leu27, Arg87, Phe25, Met128, Ala91, Ile125, Leu129, His122, and Gln85	Glu94 (2.57)
1A6	Ile140, Phe63, Met163, Arg87, Tyr126, Cys84, Ser88, Leu268, Phe81, Tyr272, Phe162, His248, Leu252, Leu251, Met147, and Leu132	No hydrogen bonds
1A8	Phe25, Leu139, Arg87, Ile140, Cys84, Val138, Leu129, Met163, Ile125, Leu132, Met128, and Ala91	Glu94 (2.57), Ser141 (3.08), and Ser88 (2.83)
1B8	Tyr126, Ile126, His122, Ser88, Leu264, Leu268, Met262, Phe159, His248, Leu252, Phe81, Tyr272, Cys84, Leu129, and Met163	No hydrogen bonds
Inflammation (COX2)		
Naproxen	Ser321, Val185, Ala495, Val317, Leu320, Phe486, Ser498, Val491, Tyr323, and Pro54	Arg89 (2.94 and 2.98)
Curcumin	Ile81, Leu61, Trp68, Phe325, Tyr84, Tyr323, Ser498, Ala495, Leu352, Tyr353, Met490, Gly494, and Ser321	Arg89 (3.00)
1A6	Trp68, Ile81, Val85, Tyr84, Leu61, Val57, Tyr323, Ser321, Leu320, Met490, Val491, Phe349, Phe486, Ser498, Trp355, Gly494, and Leu499	Arg89 (3.18)
1A8	Lys51, Phe325, Leu61, Tyr323, Trp355, Ala495, Leu320, Ser321, and Val85	Arg89 (3.05), Ser498 (2.80), and Tyr353 (2.88)
1B8	Tyr84, Val85, Leu61, Tyr323, Ser321, Leu499, Tyr316, Gly494, Phe349, Ser498, Leu320, Phe486, Met490, and Val317	Arg89 (2.970
Lipid metabolism (FAS)		
Orlistat	Ala146, Glu149, Ala150, Leu210, Tyr134, and Phe153	Arg211 (20.71 and 3.33)
Curcumin	His163, Gln156, Phe153, Leu210, Ala146, Arg135, Thr131, Tyr134, Glu149, and Ala150	Tyr130 (2.71) and Glu214 (2.59)
1A6	Phe153, Tyr130, Ser133, Tyr237, and Tyr126	No hydrogen bonds
1A8	Tyr134, Thr131, Leu210, Ile33, Tyr126, Tyr237, His256, Ser91, Tyr130, and Phe153	Tyr90 (2.86) and Asp121 (2.58)
1B8	Thr131, Lys209, Leu210, Ser91, Tyr92, Tyr130, Ala233, Gly234, Tyr126, and Gly122	No hydrogen bonds

**Table 6 ijms-25-02603-t006:** The predicted target proteins of curcumin and the three synthetic curcumin derivatives. Identified from the RSCB Protein Data Bank [89].

Protein	PDB ID	Metabolic Pathway
Proliferator-activated receptor-gamma (PPARγ)	7WGO	Adipogenesis
Mitogen-activated protein kinase 14 (MAPK14)	5XYX	
Signal transducer and activator of transcription 3 (STAT3)	6NUQ	
Acetyl-CoA carboxylase (ACC)	3TDC	Lipid metabolism
Fatty acid synthase (FAS)	2VZ8	
Glycogen synthase kinase-3 beta (GSK3β)	5F95	
Cyclooxygenase 2 (COX2)	5F1A	Inflammation
Interleukin 6 (IL6)	1ALU	
Adiponectin	6U66	

**Table 7 ijms-25-02603-t007:** Grid box co-ordinates of molecular docking for the predicted protein targets.

Target Protein	Centre (X, Y, Z)	Dimensions of Grid Box (X, Y, Z)
PPARγ	−15.2125, 9.29439, 11.01	15.7975, 13.9009, 17.6109
MAPK14	10.5915, 27.7279, 3.79848	18.5886, 14.9989, 15.8401
STAT3	5.5637, 37.1142, 12.1653	10.9664, 13.6214, 19.9643
ACC	32.2003, 61.7848, 40.5045	13.7853, 12.3731, 13.918
FASN	0.967647, 66.8543, 47.8479	19.5167, 19.0018, 20.6675
GSK3β	76.6598, −0.578815, 13.9848	17.0805, 18.7096, 17.3094
COX2	−35.499, −48.6516, −23.2359	20.5025, 29.0627, 22.0537
IL6	25.5171, −14.866, 15.8218	13.9402, 12.7375, 13.6888
Adiponectin	9.88392, 17.1713, −37.7544	14.1579, 12.833, 12.5061

## Data Availability

All additional data including raw data will be provided upon request after this paper is published.

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
