# Peer review of "Evaluating the Therapeutic Potential of Curcumin and Synthetic Derivatives: A Computational Approach to Anti-Obesity Treatments"

_ijms, 2024, doi:10.3390/ijms25052603_

Round 1

Reviewer 1 Report

Comments and Suggestions for Authors

Comments on Manuscript No: ijms-2800912

The authors in the Manuscript “Evaluating the Therapeutic Potential of Curcumin and Synthetic Derivatives: A Computational Approach to Anti-Obesity Treatments” assessed the pharmacokinetics and pharmacodynamics profiles of curcumin and synthetic curcumin derivatives using in silico approaches to identify and evaluate targeted obesity mechanisms. The computational models used as the tool to evaluate their physicochemical properties, drug likeness, toxicity, and ADME properties, including absorption through the gastrointestinal tract, distribution to target tissues, metabolism by hepatic enzymes. The interactions between these compounds and their molecular targets involved in adipogenesis, lipid metabolism, and inflammation was estimated to predict their potential efficacy in combating obesity.

Please, when the plant is first mentioned, the full Latin Name should be given – species name, author name and Family, page 2, line 59.

Page 2, line 92, please, explain what “direct organs” means?

Page 3, line 128, the Title “Compound preparation” is misleading, referring to generation of three –dimensional structures of the compounds taken for the investigation

Please, check the spelling, page 3, line 138.

Did the authors prepare the synthetic derivatives of curcumin?

Author Response

We thank the reviewer for their dedication to review our manuscript. We have attached the responses to the reviewer in a PDF document hereof.

Reviewer 2 Report

Comments and Suggestions for Authors

Comments

The authors use novel and emerging approaches, specifically computational and molecular recognition techniques, to decipher the role of curcumin and its synthetic derivatives to analyze their possible implementation as a treatment against obesity.

The topic is very interesting, it reflects the interest of the scientific community in these computational tools.

The manuscript is well written, well focused from a biological and biochemical point of view. The results are consistent with what is known about this chronic and multisystem inflammatory pathology.

The tables and figures present a well-kept design.

The results are consistent with what scientific evidence has shown. The next step will be its validation through in vitro and in vivo tests.

The discussion and conclusion are precise from a molecular and biochemical point of view and are well conducted.

Author Response

(The authors gave the same response as above.)

Reviewer 3 Report

Comments and Suggestions for Authors

This study aimed to assess the pharmacokinetic and pharmacodynamic characteristics of three synthetic derivatives of curcumin. This evaluation was conducted in comparison  to curcumin, with a specific emphasis on examining their impact on adipogenesis, inflammation, and lipid metabolism.

This paper is skillfully crafted and presents a compelling narrative. The title and abstract effectively encapsulate the primary focus of the work, providing sufficient and necessary information. The introduction furnishes background details relevant to the study, setting a solid foundation, although some information should be added. The methods are appropriately detailed, offering clarity and replicability. The authors adeptly present abundant data, ensuring alignment with the described methodology. The inclusion of pertinent figures enhances the overall comprehensibility.

The article is characterized by readability and a well-organized structure. The conclusions resonate consistently with the evidence and arguments presented. Additionally, the references are both up-to-date and comprehensive.

The paper's strengths lie in its captivating topic, a succinct and relevant introduction, a judicious selection of methodology, and a clear presentation of results through comprehensible figures. The discussion section adeptly supports the findings, substantiating them with proper references to current literature. The conclusions drawn are firmly rooted in the obtained results.

While the paper excels in various aspects, the reviewer provides constructive feedback for the authors' consideration:

1.      Please add the limitations of your study

2.      Please add the figure illustrating the scheme of the study

3.       Please add in the introduction  that both being overweight and obese have a huge impact on the development of many diseases such as metabolic dysfunction-associated steatotic liver disease (MASLD), hypertension, and stroke, PMID: 36986170, as well as has hematologic consequences [PMID: 33270290]

Author Response

(The authors gave the same response as above.)

Reviewer 4 Report

Comments and Suggestions for Authors

The manuscript entitled “Evaluating the Therapeutic Potential of Curcumin and Synthetic Derivatives: A Computational Approach to Anti-Obesity Treatments” by Marakiya T. Moetlediwa, Babalwa U. Jacka nd co-authors provides some interesting data and in my opinion could be considered by the Editor for publication in International Journal of Molecular Sciences (after minor improvements).

Minor remarks:

Please make it clear: did the authors of the manuscript prepared the curcumin derivatives 1A6, 1A8 and 1B8 or others researchers did that (for instance line 23)

Line 27: …three synthetic derivatives of curcumin as compared to the natural curcumin (?)

The following issue must be discussed: why the scientific community should have interest in curcumin derivatives: ? the natural source of curcumin is not enough to provide it to consumers? Economic issue: the synthesis of curcumin derivatives is cheaper than curcumin extraction from natural sources? Etc.

A short paragraph related to how the examined derivatives 1A6, 1A8 and 1B8 are synthetized (prepared) could be a positive point in the manuscript.

Line 235: Is a possibility to add some statistical evaluation of different response of a natural curcumin and its synthetic derivatives? Add how P values were calculated.

Table 5: there are provided predicted values or those values are true cos’ of chemical composition?

Author Response

(The authors gave the same response as above.)
